# Fear, Risk Perception, and Engagement in Preventive Behaviors for COVID-19 during Nationwide Lockdown in Nepal

**DOI:** 10.3390/vaccines11010029

**Published:** 2022-12-23

**Authors:** Ashok Khanal, Sulochan GC, Suresh Panthee, Atmika Paudel, Rakesh Ghimire, Garima Neupane, Amrit Gaire, Rukmini Sitaula, Suman Bhattarai, Shubhechchha Khadka, Bibek Khatri, Aashis Khanal, Bimala Panthee, Sharada P Wasti, Vijay S GC

**Affiliations:** 1Maharajgunj Medical Campus, Institute of Medicine, Tribhuvan University, Kathmandu 44600, Nepal; 2Active Pharmacy Pvt. Ltd., Kathmandu 44600, Nepal; 3Nepal Pharmacy Students’ Association (NPSA), Kathmandu 44600, Nepal; 4Teikyo University Institute of Medical Mycology, Otsuka 359, Hachioji 192-0352, Japan; 5Sustainable Study and Research Institute, Kathmandu 44600, Nepal; 6International Institute of Zoonosis Control, Hokkaido University, Sapporo 001-0020, Japan; 7York and Scarborough Teaching Hospitals NHS Foundation Trust, York Hospital, York YO31 8HE, UK; 8Department of Computer Science, Georgia State University, Atlanta, GA 30303, USA; 9Patan Academy of Health Sciences, School of Nursing and Midwifery, Lalitpur 44700, Nepal; 10Faculty of Education, Health & Human Sciences, University of Greenwich, London SE10 9LS, UK; 11School of Human and Health Sciences, University of Huddersfield, Huddersfield HD1 3DH, UK

**Keywords:** COVID-19, fear, health behavior, risk perception, Nepal

## Abstract

The world has faced huge negative effects from the COVID-19 pandemic between early 2020 and late 2021. Each country has implemented a range of preventive measures to minimize the risk during the COVID-19 pandemic. This study assessed the COVID-19-related fear, risk perception, and preventative behavior during the nationwide lockdown due to COVID-19 in Nepal. In a cross-sectional study, conducted in mid-2021 during the nationwide lockdown in Nepal, a total of 1484 individuals completed measures on fear of COVID-19, COVID-19 risk perception, and preventive behavior. A multiple linear regression analysis was used to identify factors associated with COVID-19 fear. The results revealed significant differences in the fear of COVID-19 in association with the perceived risk of COVID-19 and preventive behaviors. Age, risk perception, preventive behavior, and poor health status were significantly positively related to fear of COVID-19. Perceived risk and preventive behaviors uniquely predicted fear of COVID-19 over and above the effects of socio-demographic variables. Being female and unmarried were the significant factors associated with fear of COVID-19 among study respondents. Higher risk perception, poor health status, and being female were strong factors of increased fear of COVID-19. Targeted interventions are essential to integrate community-level mental health care for COVID-19 resilience.

## 1. Introduction

The magnitude of healthcare and economic burden attributable to the novel coronavirus outbreak is unparalleled globally. While there are many uncertainties regarding the future trend of the COVID-19 pandemic, as it appears, the pandemic will inevitably trigger public dilemmas, disrupt people’s daily activities, and eventually impact their health and well-being. Furthermore, in resource-limited countries with health system vulnerabilities such as Nepal [1], the government advises that the best strategy for controlling COVID-19 is prevention, primarily through strict and extended social isolation [2]. However, prolonged social isolation measures have triggered various consequences, affecting people’s everyday lives and mental health [3,4]. Previous studies have shown that the containment measures exercised to control the COVID-19 pandemic have adversely affected mental health and reported widespread fear, frustration, worry, loneliness, depression, anxiety, and stress [5,6].

One characteristic and more frequent element of a pandemic’s psychological and psychiatric effect is fear, and it must be considered and observed. Fear is an adaptive defense mechanism that mobilizes energy in the face of a potentially dangerous situation. However, it can be maladaptive if it is not well-calibrated to the real threat [7,8]. Furthermore, in a pandemic, fear elevates anxiety and stress in healthy people while intensifying the symptoms of those with pre-existing psychiatric disorders [9]. Thus, chronic or disproportionate fear associated with a highly contagious and fatal disease outbreak can significantly affect mental health. On the other hand, the fear of COVID-19 has also been shown to motivate people to engage in more healthy behaviors and to avoid risky habits, as evident from a study conducted in the US, the UK, and Germany [10]. Similarly, the perceived risk of infection was related to stringent support for public health measures [11].

However, there are still significant differences in people’s risk perception and fear of COVID-19 across different population groups and places [12,13]. As evident, most Nepali migrant returnees from India (approximately 90%) in the western region of Nepal saw COVID-19 as a low threat [14], while the majority of the participants from the eastern region (*Madhesh* province), near the Indian border, expressed worry about COVID-19 [13]. People behaving so differently during times of common challenge suggests that this pandemic’s fear, risk perception, and preventive behaviors vary significantly between different places and population groups [15].

Our study was guided by Pender’s health promotion model [16], assuming that people actively regulate their behavior in an unexpected situation such as a pandemic. Involvement in health promotion through lifestyle and behaviors enables the person to maximize their potential. However, situational influences such as fear of COVID-19, the ability to perceive risks associated with an infection, and their family members and others’ socio-demographic statuses can increase or decrease a person’s commitment to participation in health-promoting behavior, thereby determining the people’s health. Moreover, fear and risk perception are potentially strong modifiers of the epidemic evolution since they can influence new positive cases. Since engagement in protective behaviors is critical not only for preventing infection in individuals but also for limiting the spread of the virus among the community during a pandemic, therefore shielding vulnerable groups and the country.

This study aimed to assess COVID-19 risk perception, fear, and preventative behavior implementation and to explore the factors associated with fear of COVID-19 and the relationship between fear of COVID-19, risk perception, preventive behavior, and socio-demographic factors during Nepal’s national lockdown in May to June 2021 to contain the second wave of COVID-19. This will provide an understanding of the mental health implications of the fear of COVID-19 infection among Nepali adults and the efforts needed to be taken to provide adequate education and social support to decrease the viral spread, morbidity, and mortality.

## 2. Materials and Methods

### 2.1. Study Design and Participants

We conducted an online cross-sectional study during the second nationwide lockdown in Nepal. The sample for this study consisted of adults aged 18 years or above residing in Nepal. Participants infected with COVID-19 were not eligible to participate in this study.

### 2.2. Data Collection

Data were collected between May and June 2021 using Google Forms in consideration of preventing the spread of COVID-19. The survey link was made available on popular social media platforms in Nepal (i.e., Facebook, WhatsApp, and Viber). Invited participants were encouraged to forward the link to their close contacts. 

Participants’ information sheets were provided before the start of the survey. Submission of the form was considered voluntary consent to participate in the study. The privacy of the participants was considered, and identities were protected and kept confidential as each completed survey was assigned a number, with no identifiable information being related to it. Then, the data obtained were coded and saved on the researcher’s computer, and no other use was made of it. Responses were limited to one per account to avoid multiple submissions by a single respondent. The survey took approximately 10–15 minutes to complete.

### 2.3. Measures and Instruments

We developed a structured questionnaire which was pilot-tested and implemented as an anonymous survey using Google Forms. The survey questionnaire consisted of socio-demographic information and questions related to the fear, risk perception, and preventive behavior of COVID-19. The questionnaire was first developed in English. Forward and backward translation of the questionnaire was carried out to ensure language and cultural equivalence between the Nepali and English versions of the scale [17]. Furthermore, the questionnaire was pilot tested among a small sample (n = 10 adults) before the commencement of the research. It helped to refine the final version.

#### 2.3.1. Fear of COVID-19

Fear of COVID-19 was assessed using the Fear of COVID-19 Scale [7], a seven-item unidimensional self-report instrument on a five-point Likert scale (1, strongly disagree to 5, strongly agree). For each question, the minimum possible score is 1 and the maximum of 5. The possible total score ranges from 7 to 35, with higher scores indicating greater fear of COVID-19. FCV-19S is currently available in different languages, such as Japanese [18], Arabic [19], Turkish [20], Italian [21], and Romanian [22]. We used the original English version and complied with the forward and back-translation into Nepali and compared to the original English version by two bilingual researchers (A.P. and B.P.). The questionnaire was sent to an epidemiologist in Nepal to examine the difference and suitability of the questionnaire. The internal consistency or the scale’s reliability among study participants was acceptable, with a Cronbach’s alpha of 0.86. This was similar to a previous study that used FCV-19S among Nepali older adults [23].

#### 2.3.2. COVID-19 Related Risk Perception

To measure COVID-19 related risk perception, the COVID-19-related risk perceptions developed by Gerhold et al. [24] was translated by the researchers of the present study into Nepali. The content validity of the translated version was tested. This instrument consisted of two items: how likely do you think it is that (i) you might become infected with COVID-19 in the near future? (ii) people in your family and friends might become infected with COVID-19 in the future? Responses were rated on a 5-point Likert scale (1 very unlikely; 5 very likely), where a higher score reflected greater risk perception. In the previous study [24], three questions were used to measure risk perception. However, we used two statements only as a translation of the third question that had nearly the same meaning as the first question. The internal reliability index of the instrument (Cronbach’s alpha) was 0.82 showing good internal consistency.

#### 2.3.3. COVID-19 Preventive Behavior

Fourteen item preventive behaviors questionnaire was used to measure the preventive behavior against COVID-19 infection. We developed the questionnaire based on the COVID-19 prevention guidelines provided by the WHO (2020), the CDSCH (2020), and several published literatures [25,26,27,28,29] describing the preventive behaviors adopted by people in the face of the pandemic [30]. Considering that health-promotive behavior differs from person to person and according to the culture, we used a questionnaire appropriate to the Nepali context. The content of the translated questionnaire was validated in terms of its relevancy, appropriateness, adequacy, and construct towards preventive behavior in the Nepali context by content experts in the field of public health. Both the English and Nepali versions were distributed together to facilitate a clear understanding. The Cronbach’s alpha of the translated scale was 0.67, which falls within conventional value of alphas between 0.65 and 0.80, indicating adequate internal consistency [31]. Responses were recorded as ‘1, yes; 0, no’, where yes indicated appropriate behavior and no indicated inappropriate behavior. Thus, the total score ranged from 0 to 14, where higher scores indicated greater engagement in preventive behaviors.

### 2.4. Ethical Considerations

This study was performed in accordance with the Declaration of Helsinki, and approval was obtained from the Research Ethics Committee of Tribhuvan University, Institute of Medicine (Ref: 444(6-11) E 2077-078). The data were stored on a personal computer to which only the main author had access.

### 2.5. Statistical Analysis

Data were analyzed using IBM SPSS Statistics v28.0. Descriptive analysis was used to demonstrate demographic variables. For categorical data, the results were presented as frequencies and percentages, whereas continuous data were presented as mean ± standard deviation (SD). The Shapiro–Wilk test was used to test the normality of the distribution of data. For missing data, pairwise deletion was used. To examine the association between fear of COVID-19, risk perception, preventive behavior, and the participants’ demographic characteristics, an independent *t*-test and one-way ANOVA were used as appropriate.

Multiple linear regression analysis was carried out to identify the factors associated with fear of COVID-19. FCV-19S scores were categorized as low (score < 17), moderate (score 17–23), and high (score ≥ 24) based on a previous study [32]. The dependent variable used were risk perception, preventive behaviors, and FCV-19S, whereas the independent variables were the demographic characteristics of the participants (e.g., age, sex, marital status, and health status). Pearson correlation was used to explore the relationship between the study variables. A two-tailed (*p*-value < 0.05) was considered statistically significant.

## 3. Results

### 3.1. Socio-Demographic Characteristics of the Respondents

A total of 1511 responses were submitted during the study period; 1484 provided consent to participate, i.e., submitted the completed form. The mean age of respondents was 26 (SD = 7). A larger proportion (31%) of the respondents were from Bagmati province; 61% were male, 64% were single, and 77% lived with their families (Table 1). Half of the surveyed population were students (50%), followed by health workers (19%). Most of the respondents (93%) had completed their higher secondary education. Less than 10% of participants were current smokers and drinkers. Although more than 80% of the respondents reported complying with the lockdown measures, about a third (29 %) were satisfied with the governmental strategy, and more than half (57%) believed that lockdown measures could prevent COVID-19. One-fourth (24.5%) of the respondents were in isolation, and 32% had received at least one dose of the COVID vaccine. Two-thirds of the respondents (66%) had children/elderly at home (Table 1).

### 3.2. Fear of COVID-19

The FCV-19S score ranged from 7.0 to 35.0, with a mean score of 18.70 (SD = 5.66), indicating a moderate level of fear of the COVID-19 pandemic (Table 1). While using the fear categories, 37% of the study population reported low fear, 42% reported moderate fear, and 21% reported high fear. The highest mean score was found for the statement “I am most afraid of Coronavirus-19” followed by “nervousness or anxious when watching news”, indicating the role of the media in eliciting fear. Sleep problems were low, and only 15% of respondents agreed that their hands become clammy when they think about COVID-19. However, 45% of respondents agreed that they were afraid of losing their life because of COVID-19.

Almost all the independent variables used in this study have a significant difference in mean fear score except living status, current isolation status, COVID-19 vaccination status, and having children and elderly at home. The highest fear score was found among people who had poor health status. With regard to occupation, nurses had higher fear scores and laborers reported the least fear. Participants with primary education showed low fear scores compared to those in secondary or higher education. Age, risk perception, preventive behavior, and poor health were significantly correlated with COVID-19 fear (Table 2).

### 3.3. Risk Perception and Preventive Behaviors

More than one-third (36%) and about half (45%) of respondents perceived that respondents themselves or their acquaintances were infected with COVID-19, respectively. Participants assumed that the possibility of acquaintances contracting COVID-19 was higher than the possibility of themselves being infected. More than two-thirds of respondents emphasized preventive behaviors, where 95% or above practiced hand hygiene, used face masks, reduced contact with other people in presence of flu-like symptoms, got adequate sleep, and sought medical advice in the presence of COVID symptoms. However, checking body temperature, taking herbal supplements, and exercising were among the least practiced (Table 3).

### 3.4. Factors Associated with the Fear of COVID-19

To investigate which factors uniquely explained the mean difference of COVID-19 fear score, all significant continuous factors (age, risk perception, health status, and preventive behavior), and significant categorical factors were entered into the regression model. This model explained a 20.2% variation in fear of COVID-19 score (*p* < 0.001). Table 4 presents the factors associated with the fear of COVID-19 among adults. Being female, a poor health status, a high risk perception, and preventive behaviors were significant factors for the fear of COVID-19. Compared to being married, being unmarried (single or in a relationship) was inversely related to fear of COVID-19.

## 4. Discussion

This study explored the collective impact of COVID-19 during the nationwide lockdown in terms of fear, risk perception, and preventive behavior in Nepal. The mean score of the FCV-19S was 18.70, whereas 21% of respondents had high scores and 42% had moderate scores. Our findings aligned with the latest Nepal study [23] and the global literature [33]. The findings revealed that being a female, having poor health status, have a high risk perception, and showing preventive behavior were significant factors for fear of COVID-19. The recent global literature also indicated that female participants had higher levels of COVID-19 fear than their counterparts [33]. Likewise, in their review, Quadros and colleagues [34] found that more women experience moderate to high levels of COVID-19 fear. However, a recent study conducted among older adults in Nepal found similar fear levels of COVID-19 [23]. The reason behind this nonsignificant difference in older adults could be explained by choice of study sites (one province, semi-urban) and the nature of data collection. In contrast, our study participants were mostly urban-centric, and the survey was administered online. 

In our study sample, more than 40% of the participants agreed or strongly agreed that they were most afraid of COVID-19, were afraid of losing their life because of COVID-19, and became nervous and anxious when watching COVID-19 related news. This was considerably higher than a previous study among Nepalese older adults [23]. This variation could be partly explained by the fact that our study was conducted at a time when Nepal reported a record-high number of cases and deaths [35]. Amid such developments, Nepal’s healthcare system was overwhelmed, and treatment was scarce in terms of beds, medicines, oxygen supply, and ambulances.

Study participants who reported poor health status were more likely to have a relatively higher fear of COVID-19. A previous study among immunocompromised and chronic disease patients showed that more than half of the respondents agreed with the statement [35]. The most significant concerns about COVID-19 were the health of others (friends, grandparents, and loved ones), followed by healthcare collapse, economic consequences, and mass panic [8]. Although this replicates the findings from earlier studies [36,37], a higher number of deaths among people with comorbidities in Nepal could have some role to play in it [35,38].

Likewise, our study identified that government healthcare workers (HCWs) had a higher prevalence of fear, and these findings also aligned with a recently conducted meta-analysis on COVID-19 fears [33]. This might be because of continued exposure and risk of transmission to self or family, uncertain pay, and a lack of personal protective equipment in governmental hospitals [39,40]. Few earlier studies in Nepal have also reported the stigmatization of healthcare workers by community members for concern that HCWs are sources of infection, contributing to the presence of fear among health workers [41,42,43]. We also found that participants who worked as laborers or had only a primary level of education perceived the least fear of COVID-19, which could be because these people might represent daily wage earners.

Secondly, we explored the risk perception of COVID-19. One-third of the respondents reported that they and their family and friends were more likely to become infected with COVID-19. People’s willingness to adopt preventive behaviors associated with risk perception of diseases has been shown to influence the spread of COVID-19, implying that risk perception could be a key factor of pandemic evolution by influencing the new cases [44,45].

Thirdly, we explored preventive behaviors, where a large proportion of the respondents reported that they were following preventive behaviors. Our findings aligned with a recent study in Hong Kong where more than three-fourth of participants reported adhering to COVID-19-recommended preventive behaviors [46]. However, a study involving participants from Japan and Poland revealed that loneliness is negatively linked to COVID-19 preventive behaviors [44]. The majority of the participants in our study were not living alone, which partly explains the higher rate of engagement in prevention behaviors.

We found that fear was significantly and positively associated with age, risk perception, preventive behavior, and poor health status. Furthermore, higher fear and increased risk perception of getting infected was associated with higher preventive behaviors and vice versa. Even though several studies have reported that higher fear is associated with poor mental health [47,48], it is justifiable that when people realize that they are vulnerable to the risk, fear related to the disease increases, which, in turn, motivates individuals to adopt and practice preventive behaviors [49]. However, in this study, a causal relationship between preventive behavior and fear could not be made; thus, the results cannot be generalized as explained in the model. The continuous fear due to the pandemic situation is the risk of different mental illnesses in the long run. Thus, the psychosocial aspect of the people must be thoroughly considered. Community-level psychosocial awareness programs or mass media program may address community fears and continuous fear, thereby developing mental illness in the future.

### Strengths and Limitations

In our study, people with no or limited access to and literacy on the internet were underrepresented. The study sample included a large proportion of younger adults with diverse background who have access to internet services compared to older adults and have self-selected to participate in the survey. As a result, the results cannot be generalized to all the people of Nepal who are illiterate and do not have access to internet. Second, this study examined a limited number of psychological characteristics, ignoring perceived severity, perceived control, and self-efficacy. Third, our respondents might have discrepancies regarding preventive behavior. Due to the online survey, we could not control response bias. Nevertheless, this research helps in the understanding of human behavior during the pandemic and highlights different factors of COVID-19 fear and its association with risk perception and preventive behavior and can help public health experts, policymakers, and healthcare officials to implement successful public interventions assuring mental well-being to reduce the threat of pandemics. Despite these limitations, we clearly identified the factors of high fear and low fear during such pandemic situations, which sensitizes the public health managers/policy makers to take the lead for effective management of people’s mental well-being in crises. Our study included a larger proportion of younger adults due to their access to internet services, familiarity with information technology, and higher literacy.

## 5. Conclusions

This study demonstrated the fear of COVID-19 in association with the perceived risk of COVID-19 and preventive behaviors. It revealed that age, risk perception, preventive behavior, and poor health status were positively related to fear of COVID-19. Most of the study participants showed appropriate preventive practices towards COVID-19. Perceived risk and preventive behaviors uniquely predicted fear of COVID-19 over and above the effects of demographic variables. In this time of global crisis, the findings of this study might aid in identifying the groups most at risk and in developing tailored intervention methods to ensure their optimal mental well-being. Different intervention programs can be designed according to the level of fear towards the pandemic situation to reduce the traumatic consequences of COVID-19 in the long run. However, further research is needed to determine precisely the level of fear required to initiate better preventive behaviors and healthcare practices.

## Figures and Tables

**Table 1 vaccines-11-00029-t001:** Demographic distribution of the study population and comparison of mean scores of fear of COVID-19, risk perception, and preventive behavior by socio-demographic variables.

Variables	Participants n (%)	Fear of COVID-19 Score, Mean (SD)	Risk Perception Score, Mean (SD)	Preventive Behavior Score, Mean (SD)
**Gender**		***	**	*
Male	905 (61.0)	17.94 (5.79)	6.40 (1.80)	11.34 (2.18)
Female	579 (39.0)	19.89 (5.24)	6.70 (1.50)	11.63 (1.97)
**Age group**		***		
20-25 years	960 (64.7)	18.38 (5.43)	6.48 (1.59)	11.42 (2.00)
25-30 years	259 (17.5)	18.39 (5.67)	6.43 (1.97)	11.45 (2.43)
≥30 years	265 (17.9)	20.20 (6.22)	6.71 (1.74)	11.63 (2.17)
**Residence**		***	***	**
Province 1	79 (5.3)	18.95 (5.83)	6.23 (1.58)	11.11 (2.77)
Madhesh Province	177 (11.9)	19.42 (5.84)	6.32 (1.86)	11.46 (2.14)
Bagmati Province	460 (31.0)	17.75 (5.26)	6.49 (1.63)	11.23 (2.10)
Gandaki Province	141 (9.5)	19.28 (6.00)	6.48 (1.52)	11.57 (1.84)
Lumbini Province	270 (18.2)	17.79 (4.70)	6.39 (1.48)	11.60 (1.93)
Karnali Province	201 (13.5)	20.23 (6.13)	7.04 (1.81)	11.93 (2.00)
Sudurpaschim Province	156 (10.5)	19.65 (6.37)	6.49 (1.96)	11.38 (2.24)
**Occupation**		***	***	**
Healthcare worker (HCW)	283 (19.1)	19.71 (6.32)	7.42 (1.72)	11.81 (1.89)
Other government employee	125 (8.4)	18.94 (5.51)	6.38 (1.70)	11.77 (1.81)
Self-employed	157 (10.6)	18.24 (6.13)	6.35 (1.89)	11.19 (2.81)
Unemployed	89 (6.0)	20.56 (5.94)	6.39 (1.93)	11.36 (2.18)
Students	739 (49.8)	18.27 (5.29)	6.24 (1.51)	11.37 (2.08)
Others	91 (6.1)	17.75 (4.72)	6.51 (1.46)	11.23 (1.67)
**HCWs (n = 788)**		***	***	
Doctor/nurse	181 (23.0)	20.38 (6.22)	7.19 (1.76)	11.81 (1.96)
Pharmacist/Lab technician/radiologist	182 (23.1)	18.16 (5.8)	6.76 (1.72)	11.96 (1.88)
Health science students	425 (53.9)	18.52 (5.41)	6.16 (1.72)	11.68 (1.86)
**Marital status**		***	***	***
Married	422 (28.4)	20.49 (5.92)	6.96 (1.76)	11.89 (1.87)
Divorced/widowed/separated	22 (1.5)	22.05 (7.78)	7.27 (1.91)	11.64 (3.08)
Unmarried single	943 (63.5)	17.94 (5.24)	6.32 (1.59)	11.26 (2.13)
Unmarried/in a relationship	97 (6.5)	17.57 (5.98)	6.31 (1.92)	11.56 (2.31)
**Living with**				
Family	1137 (76.6)	18.84 (5.49)	6.48 (1.60)	11.46 (2.10)
Alone	211 (14.2)	18.59 (5.92)	6.60 (1.95)	11.28 (2.10)
Friends	115 (7.8)	17.52 (6.62)	6.72 (2.08)	11.88 (2.05)
Others	21 (1.4)	19.00 (5.96)	6.29 (1.45)	10.95 (2.36)
**Highest education**		*	***	
Primary	14 (0.9)	15.43 (5.84)	4.93 (2.09)	12.21 (1.67)
Secondary	91 (6.1)	19.99 (5.45)	6.48 (1.66)	11.43 (2.49)
Higher secondary	379 (25.5)	18.62 (5.60)	6.34 (1.59)	11.31 (2.02)
Bachelor and above	1000 (67.4)	18.66 (5.68)	6.60 (1.73)	11.51 (2.10)
**Monthly income (in NRs)**		**	***	
<10,000	168 (11.3)	17.84 (5.91)	6.08 (1.74)	11.15 (2.58)
10,000-20,000	278 (18.7)	19.05 (5.90)	6.47 (1.65)	11.46 (2.16)
21,000-30,000	358 (24.1)	18.13 (5.00)	6.43 (1.62)	11.39 (1.96)
31,000-40,000	290 (19.5)	19.56 (5.95)	6.67 (1.65)	11.54 (2.12)
>40,000	390 (26.3)	18.71 (5.65)	6.69 (1.77)	11.60 (1.93)
**Smoking**		**		***
Never	1260 (84.9)	18.78 (5.51)	6.51 (1.63)	11.61 (1.93)
Former	131 (8.8)	19.11 (6.07)	6.53 (1.83)	10.92 (2.76)
Current	93 (6.3)	17.00 (6.72)	6.54 (2.29)	10.28 (2.78)
**Drinking**		*	*	
Never	1150 (77.5)	18.87 (5.58)	6.51 (1.65)	11.63 (1.93)
Former	194 (13.1)	18.68 (5.93)	6.31 (1.80)	11.22 (2.41)
Current	140 (9.4)	17.33 (5.79)	6.84 (1.86)	10.40 (2.64)
**Vaccination**			***	*
Yes, single dose	145 (9.8)	17.87 (5.86)	6.83 (1.64)	11.68 (1.92)
Yes, both doses	325 (21.9)	19.64 (6.48)	6.93 (1.95)	11.68 (1.86)
No	1014(68.3)	18.52 (5.31)	6.33 (1.58)	11.36 (2.19)
**Health**		***	***	***
Very good	483 (32.5)	17.22 (5.80)	6.22 (1.82)	11.61 (2.17)
Good	719 (48.5)	19.14 (5.28)	6.66 (1.59)	11.53 (1.90)
Fair	265 (17.9)	19.82 (5.41)	6.65 (1.62)	10.99 (2.42)
Poor	17 (1.1)	24.88 (9.37)	6.41 (2.60)	11.47 (2.12)
**Suspected in family**		***	***	
Yes	429 (28.9)	19.76 (6.20)	7.21 (1.78)	11.61 (2.11)
No	914 (61.6)	18.15 (5.36)	6.15 (1.56)	11.44 (2.06)
Unsure	141 (9.5)	18.98 (5.33)	6.73 (1.55)	11.17 (2.32)
**Currently isolated**			**	***
Yes	364 (24.5)	19.00 (5.97)	6.75 (1.66)	11.75 (1.90)
No	923 (62.2)	18.66 (5.46)	6.42 (1.71)	11.44 (2.10)
Partially	197 (13.3)	18.30 (5.95)	6.52 (1.64)	11.05 (2.42)
**Follow lockdown**		***	***	***
No	32 (2.2)	12.75 (7.04)	5.06 (2.31)	8.47 (3.65)
Yes, sometimes	227 (15.3)	18.89 (5.82)	6.66 (1.62)	10.68 (2.34)
Yes, always	1225 (82.5)	18.82 (5.51)	6.52 (1.67)	11.68 (1.90)
**Satisfied with govt strategy**		*		***
Yes	425 (28.6)	19.05 (5.64)	6.45 (1.83)	12.06 (1.89)
No	676 (45.6)	18.23 (5.68)	6.44 (1.68)	11.14 (2.25)
May be	383 (25.8)	19.15 (5.60)	6.69 (1.55)	11.37 (1.93)
**Lockdown is effective**		**	***	***
Yes	840 (56.6)	19.03 (5.83)	6.62 (1.75)	11.78 (2.01)
No	146 (9.8)	17.42 (6.17)	6.05 (1.79)	10.68 (2.57)
Maybe	498 (33.6)	18.52 (5.14)	6.46 (1.54)	11.15 (2.01)
**Comorbidities**		**	***	
Yes	80 (5.4)	20.73 (7.33)	7.19 (1.73)	11.36 (2.37)
No	1404 (94.6)	18.59 (5.53)	6.47 (1.68)	11.47 (2.09)
**Children and elderly at home**				
Yes	987 (66.5)	18.80 (5.67)	6.45 (1.67)	11.53 (2.05)
No	497 (33.5)	18.50 (5.62)	6.63 (1.73)	11.34 (2.21)
Total	1484 (100)	18.7 (5.66)	6.51 (1.69)	11.46 (2.10)

One-way ANOVA, *t*-test for equality of means; * *p* < 0.05, ** *p* < 0.01, *** *p* < 0.001.

**Table 2 vaccines-11-00029-t002:** Correlation between age, risk perception, preventive behavior, and fear of COVID-19 (N = 1484).

	Age	Risk Perception	Preventive Behavior	Poor Health
Risk perception	0.029			
Preventive behavior	0.042	0.066 *		
Poor health	0.098 ***	0.094 ***	−0.087 ***	
Fear of COVID-19	0.087 ***	0.338 ***	0.179 ***	0.199 ***

* *p* < 0.05, *** *p* < 0.001.

**Table 3 vaccines-11-00029-t003:** Preventive behaviors of Nepalese adults (N = 1484).

Preventive Behavior	N (%)	
	Yes	No
Maintain distance	1187 (80.0)	297 (20.0)
Stay at home	1312 (88.4)	172 (11.6)
Use face mask	1451 (97.8)	22 (2.2)
Cover nose when coughing and sneezing	1381 (93.1)	103 (6.9)
Frequently clean your hands	1413 (95.2)	71 (4.8)
Avoid contact with other people	1431 (96.4)	53 (3.6)
Check body temperature	640 (43.1)	844 (56.9)
Avoid touching face, eyes, nose, and mouth	1088 (73.3)	396 (26.7)
Seek medical advice	1415 (95.4)	69 (4.6)
Take herbal supplements	706 (47.6)	778 (52.4)
Maintain balanced diet	1320 (88.9)	164 (11.1)
Keep informed	1339 (90.2)	145 (9.8)
Get adequate sleep	1395 (94.0)	89 (6.0)
Regularly exercise	932 (62.8)	552 (37.2)

**Table 4 vaccines-11-00029-t004:** Results of multiple regression on factors associated with the fear of COVID-19 (N = 1484).

Variables	Regression Coefficient (β)	Standard Error	95% Confidence Interval	*p*-Value
Age (years)	0.03	0.02	−0.01, 0.06	0.181
Risk perception	0.94	0.08	0.78, 1.10	<0.001
Health status	1.24	0.18	0.88, 1.60	<0.001
Preventative behavior	0.41	0.06	0.29, 0.54	<0.001
Being a female	1.38	0.27	0.85, 1.92	<0.001
Marital status—Married (Ref) Divorced/widowed/separated Unmarried—single Unmarried—in a relationship	1.28 −1.30 −1.70	1.11 0.32 0.58	−0.90, 3.45 −1.91, −0.68 −2.84, −0.56	0.251 <0.001 0.003

## Data Availability

Data are available on reasonable request from the corresponding author.

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
