# Peer review of "Fear, Risk Perception, and Engagement in Preventive Behaviors for COVID-19 during Nationwide Lockdown in Nepal"

_vaccines, 2022, doi:10.3390/vaccines11010029_

Round 1
Reviewer 1 Report
In this study authors present interesting results of a study that assessed the Covid 19 related fear, risk perception and preventive behavior during the nationwide lockdown in Nepal.
Age, risk perception, preventive behavior and poor health status were significantly positively related to fear of Covid 19.
This topic could be interesting based in the growing need to improve community level mental care for Covid 19 resilience. Howevwr we found some problems in the investigation that make the study less valid.
The survey link for data collection was made available using social platforms such as Facebook, WhatsApp and Vibe. The privacity of the participants with the use of these networks is certainly complicated. In adition to this, as the authors refre, the risk of bias is enormeously great since only people with internet acces could use it. Basically the results can not relect the real situation in the population and the study has less validity por publication
Author Response
|
Reviewer’s comments |
Authors’ reply |
|
Reviewer 1 Comments |
|
|
· In this study authors present interesting results of a study that assessed the Covid 19 related fear, risk perception and preventive behavior during the nationwide lockdown in Nepal. · Age, risk perception, preventive behavior and poor health status were significantly positively related to fear of Covid 19. · This topic could be interesting based in the growing need to improve community level mental care for Covid 19 resilience. Howevwr we found some problems in  the investigation that make the study less valid. |
Thank you for your valuable feedback. We have made suggested changes in the manuscript.
|
|
Thank you for your suggestion. We agree that sharing the survey link on social media platforms and maintaining the privacy of the survey respondents are very pertinent issues. The participant information sheet described the aims and objectives of the study and mentioned that continuing the survey would be considered informed consent to participate. The survey was designed using Google forms, and to avoid duplicate submissions, a Google account address is used as a unique identifier. Once the survey data were downloaded, the column containing the email address was deleted. So, the data do not include any personal identifiers. Furthermore, we agree that those who had access to the internet participated in the survey. As the study was conducted during the national lockdown to curtain the infection of COVID-19, it was not possible to conduct an offline (paper and pencil) survey. As apparent in other online surveys, especially people living in the urban area with higher education had to access the internet and took part in online surveys.
Besides, we also tried to obtain a snowball sampling technique to share the link among the potential respondents to lower the inherent risk of participants' privacy associated with the online survey.
We agree that the result can not be generalized to all the people of Nepal in general, as people who are illiterate and do not have internet access were not included in the study. It has been mentioned in the limitation of the study. |
|
Reviewer |
|
Reviewer 2 Report
In my opinion the paper can be informative and provide a valuable source document for anyone requiring a primer to know and understand this issue. But, numerous shortcomings in the sections Methods, Results and Disussion make this paper not appropriate for publication in this form and significant corrections should be made (major revision). Some comments:Lines 120-122: Check whether the reference No. 17 is the appropriate for the claim given in this sentence. Correct this.
Lines 123-133: State the results of the validation of this questionnaire (Fear of COVID-19) to the Nepali language, with the citation of the appropriate reference on this. Lines 131-133: State where in this manuscript is the distribution of fear of COVID-19 by level categories (low, moderate and high) shown. Lines 134-145: State the results of the validation of this questionnaire (COVID-19 Related Risk Perception) to the Nepali language, with a citation of the appropriate reference about this.
Lines 146-160: State the results of the validation of this questionnaire (COVID-19 Preventive Behavior) to the Nepali language, with a citation of the appropriate reference. Lines 175-180: Explain whether in this study, that applied the cross-sectional design, the term `predictor` could be used. Correct this in an appropriate manner. Lines 182-205: In Table 1 insert the distribution of study participants by age groups/categories. Lines 194-196: Check whether the title of Table 1 corresponds to what is presented in the table itself. Correct this. Line 195: Explain where in Table 1 is the distribution of `fear level` presented. Lines 206-213: Reconstruct the content of this paragraph and present it in a logical order. Lines 218-227: State on which of the tables are the described results presented. Lines 230-238: Explain why is age (as it stated on Line 178 in the subsection Statistical Analysis) not included in the model presented on Table 4. Line 238: For Table 4, in the legend, state the used statistical test and explain all abbreviations under the table. Lines 239-294: The section Discussion as a whole does not have a logical flow and is not comprehensive. Firstly, the variables that were described as `predictors of fear of COVID-19` were not discussed at all or not in a satisfactory way. The discussion mostly cites references from the studies that were conducted in Nepal. There are numerous similar studies conducted in other countries that should have been used in the discussion and for comparison of the results. Lines 295-308: Discuss the possible influence of `selection bias`, since the inclusion and exclusion criteria in this study enabled that a big number of responders are students (49.8%, among them 53.9% were health science students), unmarried single (63.5), that could be associated with age and level of fear.
Author Response
|
Reviewer’s comments |
Authors’ reply |
|
Reviewer 2 Comments |
|
|
In my opinion the paper can be informative and provide a valuable source document for anyone requiring a primer to know and understand this issue. But, numerous shortcomings in the sections Methods, Results and Disussion make this paper not appropriate for publication in this form and significant corrections should be made (major revision). Some comments:
|
Thank you for pointing this out. We have corrected this and inserted the following reference: Herdman, M.; Fox-Rushby, J.; Badia, X. 'Equivalence' and the translation and adaptation of health-related quality of life questionnaires. Qual Life Res 1997, 6, 237-247, doi:10.1023/a:1026410721664.
|
|
· Lines 123-133: State the results of the validation of this questionnaire (Fear of COVID-19) to the Nepali language, with the citation of the appropriate reference on this.
|
Thank you for your suggestion. The internal consistency or the reliability of the scale among Nepali older adults was acceptable (Cronbach’s α = 0.86, McDonald’s ω = 0.88 and Guttmann’s λ = 0.90).
|
|
· Lines 131-133: State where in this manuscript is the distribution of fear of COVID-19 by level categories (low, moderate and high) shown. · |
Thank you for your suggestion. This result is not presented in the manuscript at the moment. |
|
· Lines 134-145: State the results of the validation of this questionnaire (COVID-19 Related Risk Perception) to the Nepali language, with a citation of the appropriate reference about this. · |
Thank you for your suggestion. We have added the following sentence stating the results of the validation of the COVID-19 Related Risk Perception questionnaire in the Nepali language. The internal reliability index of the instrument (Cronbach’s alpha) was 0.82 showing good internal consistency. |
|
· Lines 146-160: State the results of the validation of this questionnaire (COVID-19 Preventive Behavior) to the Nepali language, with a citation of the appropriate reference.  |
Thank you for your suggestion, we have updated the Cronbach’s alpha of the translated scale was 0.70 for this study (line 175). |
|
· Lines 175-180: Explain whether in this study, that applied the cross-sectional design, the term `predictor` could be used. Correct this in an appropriate manner. · |
Thank you for your suggestion, we agree with the reviewer and have changed the term ‘predictor’ to the ‘factor associated with’ throughout the manuscript. |
|
· Lines 218-227: State on which of the tables are the described results presented. |
Thank you for your suggestion, we have referenced table 1. |
|
· Lines 182-205: In Table 1 insert the distribution of study participants by age groups/categories. · |
Thank you for your suggestion. We have inserted age categories in Table 1. As the median age was 24 years, we created 3 age categories 20-25, 25-30 and ≥30 constituting 65%, 17% and 18% of the study population, respectively. |
|
· Lines 194-196: Check whether the title of Table 1 corresponds to what is presented in the table itself. Correct this.  Line 195: Explain where in Table 1 is the distribution of `fear level` presented. · |
Thank you for pointing this out. We have corrected the title of Table 1 which now reads “Table 1. Demographic distribution of the study population and comparison of mean scores of fear of COVID-19, risk perception and preventive behaviour by socio-demographic variables”.
We have now added a sentence under section 3.2 Fear of COVID-19 to present the distribution of fear level. The revised sentence reads added text reads “While using the fear categories, 37% of the study population reported low, 42% with moderate and 21% with high fear levels, respectively.” |
|
· Lines 206-213: Reconstruct the content of this paragraph and present it in a logical order.   · |
Thank you for your suggestion. As per the suggestion we have reconstructed the content of the paragraph as follows:
Almost all the independent variables used in this study have a significant difference in mean fear score except living status, current isolation status, COVID-19 vaccination status and having children and elderly at home. The highest fear score was found among the people who had poor health status. With regards to the occupation, nurses had higher fear score, and laborers reported the least fear. Participants with primary education showed low fear score compared to those in secondary or higher education. Age, risk perception, preventive behavior and poor health were significantly correlated with COVID-19 fear (Table 2). |
|
· Lines 230-238: Explain why is age (as it stated on Line 178 in the subsection Statistical Analysis) not included in the model presented on Table 4. |
Thank you for your suggestion. We regret the omission of age from the regression analysis. Indeed, age was included in the regression model. We have updated Table 4. |
|
· Line 238: For Table 4, in the legend, state the used statistical test and explain all abbreviations under the table. · |
Thank you for your suggestion, we have now simplified the table and report regression coefficients, standard error, 95% confidence interval and p-value to make the table more readable. We have also changed the title of Table 4 to reflect this which now reads “Table 4. Results of multiple regression on factors associated with the fear of COVID-19 (N=1484).” As the table title is self-explanatory and it doesn’t include abbreviations, a table footnote was not needed. |
|
· Lines 239-294: The section Discussion as a whole does not have a logical flow and is not comprehensive. Firstly, the variables that were described as `predictors of fear of COVID-19` were not discussed at all or not in a satisfactory way. |
Thank you for your suggestion. We have now revised the discussion section.
|
|
· The discussion mostly cites references from the studies that were conducted in Nepal. There are numerous similar studies conducted in other countries that should have been used in the discussion and for comparison of the results. |
Thank you for your suggestion. We have cited studies conducted other than Nepal and compared and contrasted the results. |
|
· Lines 295-308: Discuss the possible influence of `selection bias`, since the inclusion and exclusion criteria in this study enabled that a big number of responders are students (49.8%, among them 53.9% were health science students), unmarried single (63.5), that could be associated with age and level of fear. |
Thank you for your suggestion. We have updated the discussion section. |
Reviewer 3 Report
The manuscript “Fear, Risk Perception, and Engagement in Preventive Behaviors for COVID-19 During Nationwide Lockdown in Nepal” assessed the COVID-19 related fear, risk perception and preventative behavior. The study is important as targeted intervention could be designed accordingly, to initiate better preventive behaviors and healthcare practices.
The work is interesting and precedent, although it is preliminary.
I think it is acceptable after some revision, taking into account the following points.
Major points:
1. Line 245 “Our finding is considerably higher than a previous study among Nepalese older adults [31]. This variation could be partly explained by the fact that our study was conducted at a time when Nepal reported a record high number of cases and deaths [32].” The authors referred to a prior study among older adults (age), yet partly explained it with that study conducted during high number of cases (time). The reasoning could be improved (prior study’s age vs time). If the authors refer to the mean age of respondents in their own study, (prior study’s age vs current study’s age), the reasoning might be improved.
2. Line 206 “We further found that age, risk perception, preventive behavior and poor health were significantly positively related to fear”. It will help the audience to better understand the content, if age distribution is clearly demonstrated, for the people in different age group (For example, 20-30, 30-40, 40-50,…).
Minor points:
3. In section 4.1, the authors demonstrated some limitations of the study. Have the authors considered the age’s effect on online behavior and response rate to survey, which may lead to underrepresentation?
4. Line 338. Acknowledgements section “In this section, you can acknowledge…”. These sentences from template need to be updated. Please write this section accordingly.
Author Response
|
Reviewer’s comments |
Authors’ reply |
|
Reviewer 3 Comments |
|
|
The manuscript “Fear, Risk Perception, and Engagement in Preventive Behaviors for COVID-19 During Nationwide Lockdown in Nepal” assessed the COVID-19 related fear, risk perception and preventative behavior. The study is important as targeted intervention could be designed accordingly, to initiate better preventive behaviors and healthcare practices.  The work is interesting and precedent, although it is preliminary.  I think it is acceptable after some revision, taking into account the following points.  |
Thank you very much for your positive feedback. 
|
|
    Line 245 “Our finding is considerably higher than a previous study among Nepalese older adults [31]. This variation could be partly explained by the fact that our study was conducted at a time when Nepal reported a record high number of cases and deaths [32].” The authors referred to a prior study among older adults (age), yet partly explained it with that study conducted during high number of cases (time). The reasoning could be improved (prior study’s age vs time). If the authors refer to the mean age of respondents in their own study,  (prior study’s age vs current study’s age), the reasoning might be improved.  |
Thank you for your suggestion. In line with Reviewer #2’s suggestion, we have updated the discussion and also cited international literature. |
|
    Line 206 “We further found that age, risk perception, preventive behavior and poor health were significantly positively related to fear”. It will help the audience to better understand the content, if age distribution is clearly demonstrated, for the people in different age group (For example, 20-30, 30-40, 40-50,…).  |
Thank you for your suggestion. We’ve included age categories and reported results in Table 1.
|
|
In section 4.1, the authors demonstrated some limitations of the study. Have the authors considered the age’s effect on online behavior and response rate to survey, which may lead to underrepresentation?  |
Thank you for your suggestion. We have mentioned the large proportion of the study population were young adults and had access to internet services, and also highlighted the generalisability of study findings to other population group.
|
|
   Line 338. Acknowledgements section “In this section, you can acknowledge…”. These sentences from template need to be updated. Please write this section accordingly. |
Thank you for your suggestion, we have updated the acknowledgement section. 
|
Round 2
Reviewer 1 Report
I belive the article would be acceptable for publication including the comments made after the review
Reviewer 2 Report
I would like to thank the authors for carefully revising their manuscript, acknowledging the remarks I've given and incorporating the necessary changes in line with my comments. I believe the changes authors included in the revised manuscript made their work more comprehensive, more readable and coherent.